

# Isotopic and chromatographic fingerprinting of the sources of dissolved organic carbon in a shallow coastal aquifer

Karina T. Meredith[1,2], Andy Baker[2,4], Martin S. Andersen[2,3], Denis M. O'Carroll[2,3], Helen Rutlidge[2,3], Liza K. McDonough[2,4], Phetdala Oudone[2,4], Eliza Bryan[5], Nur Syahiza Zainuddin[6].

[1]Australian Nuclear Science and Technology Organisation, Lucas Heights, NSW, 2234, Australia.
[2]Connected Waters Initiative Research Centre, UNSW Sydney, Australia.
[3]School of Civil and Environmental Engineering, UNSW Sydney, Australia.
[4]School of Biological, Earth and Environmental Sciences, UNSW Sydney
[5]Golder Associates, Sydney, NSW, Australia. 124 Pacific Highway, St. Leonards, New South Wales 2065, Australia.
[6]Faculty of Civil Engineering, Universiti Teknologi MARA, Shah Alam, Selangor, Malaysia.

Correspondence to: Karina T. Meredith[1] (kmj@ansto.gov.au)

**Abstract.** The terrestrial sub-surface is the largest source of freshwater globally. The organic carbon contained within it and processes controlling its concentration remain largely unknown. The global median concentration of dissolved organic carbon in groundwater is low compared to surface waters suggesting significant processing in the subsurface. Yet the processes that remove this dissolved organic carbon (DOC) in groundwater are not fully understood. The purpose of this study was to investigate the different sources and processes influencing DOC in a shallow anoxic coastal aquifer. Uniquely, this study combines liquid chromatography organic carbon detection with inorganic ($\delta^{13}C_{DIC}$) and organic ($\delta^{13}C_{DOC}$) carbon isotope geochemical analyses, to fingerprint the various DOC sources that influence the concentration, carbon isotopic composition and character with distance from surface water sources, depth below surface and groundwater residence time (using $^3H$) in groundwater. It was found that the average groundwater DOC concentration was five times higher (5 mg L$^{-1}$) than the global median concentration and it doubled with depth, but the chromatographic character did not change significantly. The anoxic saturated conditions of the aquifer have limited the rate of organic matter processing leading to enhanced preservation and storage of the sources such as peats and palaeosols. All groundwater samples are more aromatic for their molecular weight in comparison to lakes and rivers and surface marine samples. The destabilisation or changes in hydrology, whether by anthropogenic or natural processes could lead to the flux of up to ten times more unreacted organic carbon from this coastal aquifer than compared to deeper inland aquifers.

## 1 Introduction

Organic matter (OM) in aquatic systems forms a fundamental part of the global carbon cycle. The soil and unsaturated zone has the potential to store at least three times more organic carbon than in the atmosphere or in living plants (Schmidt et al., 2011, Fontaine et al., 2007) and represents the largest source of carbon within terrestrial ecosystems (Keiluweit, et al., 2017).



The below-ground environment forms the largest source of freshwater and the organic carbon within these systems remains largely unaccounted for in the global carbon budget. Groundwater replenishment occurs as either diffuse recharge through the OM-rich soil zone and/or as direct recharge originating from streams and wetlands that have the potential to contain high concentrations of OM. Despite the many sources of OM, groundwater DOC concentrations are typically low (~1 mg L$^{-1}$ for

the global median DOC concentration; McDonough et al., (2019)), suggesting that significant processing has occurred in the subsurface.

Very few studies have investigated why low concentrations of DOC in groundwater occur. Recent research has provided evidence of the attenuation of chromophoric dissolved organic matter (DOM) at a scale of tens of meters along a flow path

through measurements of increased optical clarity of groundwater (Chapelle et al., 2016). This attenuation was proposed to be due to the combined effects of biodegradation and sorption. Chapelle et al. (2012) also showed that the presence of a hyperbolic relationship between DOC concentrations and dissolved oxygen provides indirect evidence for groundwater DOC that is bioavailable to microbes. Direct monitoring of groundwater DOC within a fractured rock aquifer in South Carolina, USA, including measurements of both bioavailable and chromophoric DOM were performed by Shen et al., 2015. They

demonstrated a decrease in lignin-derived phenols in groundwater compared to surface inputs, and prevalent amino acids in groundwater. At this site, it was proposed that a small fraction (8±4%) of groundwater DOC was bioavailable and that a substantial fraction of groundwater DOC was of bacterial origin (15-34%). Shen et al. (2015) suggested that DOM mobility can be described by a 'Regional Chromatography Model' (Hedges et al., 1986, 1994) as it moves through the soil column to the groundwater. It is shown that the molecular size, polarity, charge and bioavailability determine the observed decrease in

hydrophobicity of DOM along the flow path due to their controls on sorption, desorption, biodegradation and biosynthesis (Shen et al., 2015).

Carbon isotope ratios were first used in the 1960s to distinguish sources of OM in the coastal zone to identify the difference between saltmarsh sediments and freshwater peats (Emery et al., 1967). The use of carbon isotopes as tracers has been

instrumental in providing greater understandings of the sources of carbon in coastal, terrestrial and marine environments



(Berner et al., 1987, Lamb et al., 2006). Very few studies that use DOM chromatographic techniques have employed isotopic techniques to understand the source of OM in groundwater, and vice versa. Use of multiple DOM characterisation techniques to improve our understanding of the role of different OM sources in contributing to the concentration, character, and its subsequent processing along groundwater flow paths is therefore in its early stages of research.

Further work is clearly warranted to improve our understanding of the role of different OM sources and the physicochemical properties of aquifers on groundwater DOM. To that end, we present groundwater DOC concentration, isotopic signatures and organic character data from a shallow (less than 20 in depth) anoxic coastal aquifer. Within this hydrogeologically well characterised system, it is anticipated that multiple sources of OM exist including wetland, soil, peat and palaeosols. To

10 better understand and test a regional chromatography model, we utilise a chromatographic technique (liquid chromatography organic carbon detection- LC-OCD; Huber et al. (2011)). Uniquely, this study combines the chromatographic technique with inorganic and organic isotope geochemical analyses, to fingerprint the various DOC sources that influence the concentration, carbon isotopic composition ($\delta^{13}C_{DOC}$) and character with distance, depth and groundwater residence time (using $^3H$). This is done to identify controls on groundwater DOC sources and processing in a coastal groundwater system.

**2 Environmental Setting**

The coastal environment of Samurai Beach is located north-east of Anna Bay, New South Wales, Australia (Fig. 1). The Holocene sand dunes at the site rise 30 m above sea level and extend up to 800 m inland. A freshwater wetland and forest lie in the northwest corner of the sand dunes near Site 1 (i.e., S1) and the aquifer is part of the Tomaree Groundwater Source supplying towns in the Hunter Valley with portable water. The local geology of the field site was investigated during

borehole construction and using hydraulic profiling tools to produce the lithology cross-section depicted in Fig. 2 (Maric, 2013 and Howley, 2014). The lithological cross-section is parallel with the main groundwater flow direction from the wetland to the coast.

The upper 15 m of the aquifer contains a combination of barrier and back barrier sand deposits, capped by aeolian dune deposits. Site 1 (S1) is located in the northwest corner, closest to the freshwater wetland, while Site 5 (S5) is closest to the





Pacific Ocean. The lithology of the aquifer differs with distance along the transect, with a peat layer identified at ~5 m below ground level at Sites 1 and 2, and a clay unit at Site 5 (Fig 2). A shallower organic-rich layer was also identified at about 1 m bgl (below ground level) at Site 1.

Five sites were drilled and wells installed to three different depths; shallow (S_3.4 to 5.0 m bgl), medium (M_9.8 to 12.5 m) and deep (D_12.7 to 17.5 m) forming a ~500 m transect. The boreholes were drilled by hollow stem auger using a Geoprobe rig and wells were constructed out of 50 mm diameter PVC with screened intervals of 1 m located at the bottom.

Three different sources of groundwater were identified based on hydrochemistry and the evaluation of groundwater flow direction (Maric, 2013 and Howley, 2014). These groundwater sources include; (1) direct rainfall recharge through the dunes, (2) indirect wetland infiltration, and (3) deeper regional groundwater (Fig. 2). Multiple sources of OM maybe present at the site such as from (1) surface vegetation (2) transported particulate organic matter (POM) and/or (3) in-situ sedimentary sources such as peat, palaeosol or finely disseminated POM.

## 3 Methods

From 17$^{th}$ to 22$^{nd}$ of February 2014, fifteen groundwater samples were collected from the five nested sites (Fig. 2). Surface water samples were also collected from the adjacent wetland. Waters were measured for major ion chemistry, carbon isotopes ($\delta^{13}C_{DIC}$, $\delta^{13}C_{DOC}$ and $^{14}C_{DIC}$), tritium ($^{3}H$) and DOM character.

Groundwater samples were generally collected using a submersible centrifugal pump (Monsoon). A HACH multimeter (HQ40d) and probes were used for Dissolve Oxygen (LDO probe) and pH measurements in an inline flow-cell (Wattera) isolated from the atmosphere. Sub-samples for laboratory analysis we collected through an in-line, 0.45 μm filter (bypassing the flow-cell), with $\delta^{13}C_{DIC}$ and $\delta^{13}C_{DOC}$ samples further filtered through 0.22 μm. Total alkalinity was determined in the field by Gran-titration (Stumm and Morgan, 1996) using a HACH Digital Titrator (Model 16900) and 0.16 N $H_2SO_4$. Samples for anions were collected in 50 mL Polyethylene centrifuge vials bottles, with no further treatment, but stored cool (~5 °C) and dark. Samples for cations were collected in 20 mL HDPE acid-washed bottles and acidified with 1% of concentrated nitric acid ($HNO_3$). The $\delta^{13}C_{DIC}$ samples were collected in pre-combusted 12 mL glass vials (Exetainers) with no head space. The DOC and $\delta^{13}C_{DOC}$ were collected in acid washed 60 mL HDPE bottles and frozen within 12 hours of collection. Samples for

[14]C and [3]H analysis were collected in 1 L Nalgene HDPE bottles and were sealed with tape to eliminate atmospheric exchange during storage. Major and minor cations were determined using a Perkin Elmer NexION300D ICP-MS and Perkin Elmer Optima 7300 ICP-OES. Anions (for Cl and $SO_4$) were analysed using Dionex IC1000 Ion Chromatography System. Cations and anions were assessed for accuracy by evaluating the charge balance error percentage (CBE%; Table 2). Samples

fell within the acceptable ±5% range.

The $\delta^{13}C_{DIC}$ isotopic ratios of waters were analysed by Isotope Ratio Mass Spectrometer and results were reported as ‰ deviation from the international carbonate standard, NBS19 with a precision of ±0.1‰ according to methods reported in Meredith et al., (2016). The DOC concentration and $\delta^{13}C_{DOC}$ were analysed using a total organic carbon analyser interfaced to a PDZ Europa20-20 IRMS utilising a GD-100 gas trap interface. Results were reported as per mil (‰) deviation from the

NIST standard reference material with an analytical precision of ±0.6‰. The [3]H activities were expressed in tritium units (TU) with an uncertainty of ±0.1 TU and quantification limit of 0.3 TU. Samples were analysed by liquid scintillation counting. Extended methods for [3]H activities can be found in Meredith et al., (2012).

The dissolved organic matter (DOM) character was determined using optical spectroscopy and liquid chromatography (LC-OCD). The LC-OCD technique is a size-exclusion chromatographic technique that allows for the characterisation of DOC,

based on molecular weight, into six fractions. The fractions obtained are biopolymers (> 20 kDa), humic substances (~1000 Da), building blocks (300-500 Da), low molecular weight acids (< 350 Da), and low molecular weight neutrals (< 350 Da) and a hydrophobic fraction (fraction of DOC that remains in the column and determined by the difference between total DOC and the total of the other fractions), for full details see Huber et al., (2011). The humic substances fraction is further characterised for its molecular weight (based on retention time of the humic substances peak) and aromaticity (the specific

UV absorption at 254nm of the humic substances peak). Calibration is based on the chromatograms of the International Humic Substances Society (IHSS) Suwanee River humic and fulvic acid standards.

## 4 Results

Although common at coastal sites, a fresh-saline water interface was not identified in the wells, even though Site 5 is located ~100 m from the ocean. Groundwater had low salinities with Cl concentrations ranging from 0.5-1.2 mmol L$^{-1}$. The highest



Cl concentration occurred in the deeper groundwater at Sites 1 and 2 (Table 1 and Fig. 3a). Water level data suggested that under most conditions groundwater follows a west to east direction towards the coastline (Fig. 2). Notably, groundwaters are anoxic with dissolved oxygen below 0.2 mg L$^{-1}$ (Table 1). Water level patterns suggest that a buried-peat layer identified at Sites 1 and 2 is restricting the vertical flow of rainfall derived recharge into the deeper sections of the aquifer at these two

sites.

The presence of detectable $^3$H in all samples (>0.7 TU) indicated groundwater has a component of water that has been in contact with the atmosphere during the past decade. Variations in $^3$H contents were seen with depth and distance from the wetland suggesting groundwaters have varied water residence times (Fig. 3b). The wetland had the highest $^3$H content (1.7 TU) and represented a rainfall value for the region (Tadros et al., 2014). All wells located between 3.4 to 12.5 m bgl have

high $^3$H contents greater than 1.2 TU. Groundwaters from Site 3 have consistent values around 1.5 TU, suggesting a similar source of water and that the aquifer is hydraulically connected at this site. Deeper groundwaters (other than Site 3) have lower $^3$H values (less than 1 T.U.), indicating slightly older groundwater at depth (Table 2).

The $\delta^{13}C_{DIC}$ values ranged from -14.8 (S3_S) to -2.6 ‰ (S2_D) with an average of -9.7 ‰ (n = 15). The wetland had a

significantly lower $\delta^{13}C_{DIC}$ value (-23.6 ‰).The higher $\delta^{13}C_{DIC}$ values found in the deeper system at Sites 1 and 2 (-4.1 to -2.6 ‰) could indicate marine carbonate dissolution with the groundwater isotopic signature moving closer towards 0 ‰. But interestingly the $^{14}C_{DIC}$ values measured on these samples were very high (>98 pMC) indicating a modern source of inorganic carbon. The low Ca concentrations in the deeper waters at Sites 1 and 2 could suggest ion exchange processes or that carbonate dissolution is not a dominant process along flow paths leading to these sites. Ionic ratios of Na/Cl range from

0.7 to 1.3 with an average of 0.9, which is close to unity, suggesting that ion exchange between Ca and Na is not significantly influencing the concentration of these ions. Furthermore, the lack of detectable nitrate and sulphate together with increased dissolved ferrous iron and ammonia concentrations with depth, suggests strongly anoxic conditions where methanogensis is likely to occur (Table 1). The process of methanogenesis could also transform OM to DIC. Methanogensis in this anoxic environment would also explain the enrichment trend in $\delta^{13}C_{DIC}$ in the deeper waters particularly at Sites 1 and

2, however our results do not provide conclusive evidence of this process.





The average DOC concentration in groundwater for this site is high (5.0 mg L$^{-1}$; n = 15). The carbon isotopic signature of the groundwater DOC represents a C$_3$ vegetation signature (average $\delta^{13}$C$_{DOC}$ value of -27.4 ‰; n = 15). Significant variation in DOC concentration and $\delta^{13}$C$_{DOC}$ values occur within the aquifer (Fig. 4). The increase in DOC concentration in deeper groundwaters compared to shallow (except Site 4) suggests a source(s) of DOC within the aquifer, which is generally not seen in groundwater environments and will be discussed below.

The LC-OCD results did not show significant differences in DOM character in the groundwater (Fig. 5). The humic substances fraction was generally between 40-60% of the total DOC (Fig. 5a). Minor exceptions existed in the shallow samples at Sites 2 and 3 for humic substances and at Sites 2, 3 and 5 for the Low Molecular Weight-Neutrals (LMW-Ns). The LMW-Ns were low representing approximately 10% of the total DOM and they did not change significantly along the groundwater flow path (i.e. with distance from the wetland) (Fig. 5b).

**4.1 DOC variation in groundwater**

The wetland had the highest DOC (18 mg L$^{-1}$) concentration and the lowest carbon isotope value ($\delta^{13}$C$_{DOC}$ = -30.3 ‰) for the site. It also had a similar LC-OCD character to groundwater. The closest groundwater sample to the wetland located less than 5 metres downgradient at 3.5 depth (S1_S) had a significantly lower DOC concentration (5 mg L$^{-1}$) and 1 ‰ higher carbon isotopes value ($\delta^{13}$C$_{DOC}$ = -29.3 ‰). Interestingly this decrease in DOC is not reflected in changes in the mass fraction of the LC-OCD results.

The deeper samples (>17.0 m) at Sites 1 and 2 have the highest DOC concentrations (10 ± 0.5 mg L$^{-1}$) for groundwaters at the site. The $\delta^{13}$C$_{DOC}$ values are higher in the deeper groundwaters (-27.6 ± 0.2 ‰) compared to the shallow samples (-29.0 ± 0.2 ‰). The humic substances aromaticity and molecular weight shows there are two distinct groups of DOM (Fig 6). Samples that generally have lower values are from Sites 1, 2 and the wetland. All samples are more aromatic for their molecular weight in comparison to surface lakes and rivers and marine samples (Huber et al., 2011).

Groundwaters from Sites 3 to 5 had lower DOC concentrations (<6 ppm) and a 1 ‰ higher average $\delta^{13}$C$_{DOC}$ value (-26.6 ‰) than shallow and medium groundwater samples from Sites 1 and 2. Samples from S3, S4 and S5_S also had higher humic substances aromaticity and humic substances molecular weight (Fig. 6) Importantly, DOC would be expected to have decreased aromaticity and molecular weight along a flow line, the opposite to what is observed here if the samples formed a





degradation pathway. This suggests that the OM source in this section of the aquifer differs from Sites 1 and 2. Sample S5_S is the exception where it has the highest $\delta^{13}C_{DOC}$ value, which falls within the marine OM range (Fig. 4b).

## 5 Discussion

Our results show that groundwater DOC varies in concentration and isotopic character along a 500 m groundwater flow path.

The average groundwater DOC concentration found in this coastal site is five times higher (5 mg L$^{-1}$) than the global median DOC concentration for groundwaters (McDonough et al., 2019). Interestingly, we also see that the concentration of DOC doubles with depth, but we do not see any consistent trends of changing DOM chromatographic character related to depth (Fig. 5). These are in contrast to results from a deeper fractured rock aquifer (Shen et al., 2015), where DOC decreased with depth. Furthermore, the DOM character does not change significantly along the groundwater flow path, contrary to what was

found in other studies where biodegradation, sorption, desorption and biosynthesis controlled DOM (Chappelle et al., 2012; Shen et al., 2015).

It is well known that OM is more readily preserved under anoxic conditions (Bertrand and Lallier-Verges, 1993), particularly in saturated environments. In such systems, remineralisation can be low, leading to enhanced preservation and storage of OM (Schefuß et al., 2016). In fact, marine sediments, wetlands or peatlands have been suggested to have between 60-95%

reduced mineralisation rates (Keiluweit, et al., 2017). The OM itself is thermodynamically unstable but Schmidt et al., (2011) suggests it can persist because of the physicochemical and potentially biological influences of the surrounding environment that reduces the rate of decomposition.

Our hydrochemical data shows there is strong evidence to suggest OM degradation has occurred. Dissolved oxygen is less than 0.2 mg L$^{-1}$ for all groundwaters, which we infer to be due to the respiration of microbes that are adapted to access OM

in relatively anoxic conditions. Additional evidence for microbial activity includes the presence of significant concentrations of reduced redox-sensitive species such as ammonium (up to 0.9 mg L$^{-1}$) and ferrous iron (up to 3.3 mg L$^{-1}$) together with very low nitrate concentration (<0.3 mg L$^{-1}$), the absence of sulphate in the deeper groundwaters (Table 1) and detection of H$_2$S odour. In anoxic subsurface environments microbes utilise nitrate, Fe-oxides and sulphate as electron acceptors in the absence of oxygen increasing the ammonia, ferrous iron and sulphide concentrations (Berner, 1981; Appelo and Postma,

2005). A possible explanation for the constant character of the DOM with the evolved inorganic redox chemistry is that all DOM fractions are being released from the OM sources at constant rates, combined with a slightly slower decomposition of all dissolved fractions.

### 5.1 Sources of OM

The wetland Organic-rich sediments such as organic muds and silts associated with freshwater depressions are characteristic of coastal dune-slack systems which have formed since the maximum Holocene sea-level transgression and have a global occurrence, including along the southeastern coast of Australia (MacPhail, 1973). These units have the potential to form peat. In suitable environments, the decay of OM under anaerobic conditions is much slower (Berner et al., 1984a), allowing a greater accumulation of refractory OM, which may lead to peat accumulation (Lamb et al., 2006). Three 'peat' horizons

were identified in a similar coastal environment to Anna Bay (Finga Bay, Central Coast, NSW) and ranged in age from 3.0 to 6.5 ka (MacPhail, 1973). The peat formation was suggested to be very rapid, with metres of sediment accumulating over the past few thousand years.

The sand dunes at Anna Bay are likely to have formed around 7 ka (Sloss et al., 2007, Jones 1990, MacPhail, 1973) and the OM-rich layers contained within them would have formed over this time. OM can be characterised by very long turnover

15  times that increase with depth from 1 to 10 ka depending on the system (Schmidt et al., 2011). The destabilisation of these older OM-rich units whether by anthropogenic or natural processes could result in an increase in the flux of older carbon into the surface water environment (Moore et al., 2013). Furthermore the transport of this carbon has been shown to be dependent on the hydrological response in coastal systems (Webb et al., 2018).

### 5.1.1 Wetland OM

Initially, it was thought that the wetland was the major source of OM for the groundwater system because it contained elevated DOC concentrations (18 mg L$^{-1}$) and it appeared to be hydraulically connected to the aquifer. Based on these assumptions, it would mean that approximately 70% of the OM from the wetland is removed after groundwater recharge (i.e. 10 metres downgradient at sample S1_S at 3.5 m depth). This estimate seems reasonable when considering Shen et al., (2015) found that about 90% of surface-derived DOC was removed prior to reaching the saturated zone.





Our data shows that the chromatographic character of the DOM in the wetland is similar to the groundwater, suggesting if the wetland was a significant source of OM, the chemical composition and bioavailability of the wetland derived DOM remains relatively unchanged during transport through the aquifer. The higher relative contribution of DOC and evidence for the limited transformation of the wetland OM at this site could be explained by the limited sorption capacity of the

predominantly quartz sand aquifer.

If biological processing was influencing the wetland DOM during transport into the groundwater, it would be expected that the $\delta^{13}C_{DOC}$ values would become lighter than the original $C_3$ vegetation source (Benner et al., 1987). However, the carbon isotope value ($\delta^{13}C_{DOC}$) of the OM in the wetland is 1 ‰ lower compared to the shallowest groundwater sample located near the wetland. This suggests that if the wetland was the source of OM then the carbon isotopes were fractionated after

recharge.

Alternatively, the DOM in the groundwater system has a different source to the wetland. A different source of OM would also explain the distinctly different water chemistry of the wetland sample compared to the groundwater (Table 1). The wetland contains elevated levels of DOC but it is likely that the OM is being mobilised and deposited into the hyporheic zone of the wetland (i.e. 1-2 metres) by either sorption or abiotic transformation (Kerner et al., 2003). This hypothesis is

being investigated with further detailed water-sediment investigations at this location. Significantly this study does show that the modern wetland is not a major source of DOM for these groundwaters.

### 5.1.2 In-situ OM sources

Organic-rich sediments are present in the unconsolidated aquifer units and it is clear that the physicochemical properties of the aquifer are governing the persistence of the OM within the aquifer. We see there are several *in-situ* DOM sources based

on the carbon isotopes and the humic substances aromaticity and molecular weight (Fig. 6). Two major groundwater groups were identified at Sites 1-2 and Sites 3-5. The aromaticity of the humic substances in both groups is higher than those previously reported for rivers, lakes for the corresponding molecular weight, especially for Sites 3-5.

Furthermore, we see groupings for the deeper samples from Sites 1 and 2 that corresponds with carbon isotopes that are 1 ‰ higher (-27.6 ± 0.2 ‰) than other groundwater at Site 1 and 2. These groundwaters are older (<0.8 T.U.) and the OM is

likely to have originated from a deeper older palaeosol unit contained within the coastal sediments (Fig. 7). The shallow



Hydrology and
groundwaters have lower DOC concentrations and lower carbon isotope values (-29.0 ± 0.2 ‰) suggesting the OM originating from these sources are also different from the deeper palaeosol. The overlying peat units most likely formed in a similar environment to the palaeosol located at depth but contain younger OM with lower $\delta^{13}C_{DOC}$ values.

Groundwater from Sites 3-5 have lower DOC concentrations (<6 mg L$^{-1}$) and higher humic substances aromaticity and

humic substances molecular weight (Fig. 6) together with higher consistent $\delta^{13}C_{DOC}$ values (average $\delta^{13}C_{DOC}$; -26.6 ‰). The exception is the shallow sample at Site 5, which has the highest $\delta^{13}C_{DOC}$ value (-25.0‰) and trends towards a marine OM value (Lamb et al., 2006) with a DOC concentration well below the average (2.1 mg L$^{-1}$) further suggesting a localised source of OM that may be derived from marine sources due to its close proximity to the ocean. The stability and similarity in the overall chromatographic and isotopic character suggests that these groundwaters contain OM from within the aquifer that

has not undergone significant processing.

### 5.2 Implications

Most studies investigate the character or carbon isotope signatures for DOC in groundwater but rarely use both techniques. Our findings show that there are several sources of OM ranging from buried peat units to palaeosols, and without utilising both techniques these sources would not have been identified. This is important to understand the sources for estimating the

contribution of carbon to the global carbon cycle. The combination of the low sorption capacity of the coastal aquifer sediments, the presence of various sources of OM, together with associated anoxic conditions, appears to have limited the sorption and/or biodegradation that might be observed elsewhere in groundwater. Therefore the persistence of OM is found to be due to complex interactions between the OM and its environment as suggested by Schmidt et al. (2011).

The groundwater that discharges from the young unconsolidated coastal environments such as this have much higher DOC

concentrations than groundwater from older deeper aquifers that have undergone a greater degree of OM processing. The degradative processes that act on freshly produced DOM can produce greenhouse gases and produce less reactive DOM that is exported (Davidson and Janssens, 2006; Zhou, et al., 2018). This means that the DOM transported from this coastal system has the potential to impact the carbon budget. Moreover, sandy coastal aquifers containing palaeosol horizons are globally widespread as they formed during Holocene sea-level changes. These systems are sensitive to sea level changes and

climate change drivers and are likely to affect DOC export because of the changes in either hydrology or ecosystem





dynamics. The OM sources found in this system are currently saturated and anoxic. If the physicochemical conditions of the aquifer were altered, these coastal groundwater systems would then have the potential to export an order of magnitude higher volume of unreacted carbon to the surface than previously realised. These coastal groundwater systems form a significant source that is unaccounted for in the global carbon budget.

## 6 Conclusion

The purpose of this study was to investigate the role of different OM sources and the influence the physicochemical properties of an aquifer has on groundwater DOM. This was done to identify the major controls on groundwater DOC sources and processing in a coastal groundwater system. Our results showed that groundwater DOC varied in concentration and isotopic character along a 500 m groundwater flow path. The average groundwater DOC concentration was five times higher (5 mg L$^{-1}$) than the global median DOC concentration for groundwaters. The concentration of DOC doubled with depth, but the DOM chromatographic character did not change significantly with depth or along the groundwater flow path.

Multiple sources of organic matter were identified by measuring the concentration, carbon isotopic composition ($\delta^{13}C_{DOC}$) and character (by LC-OCD) of groundwater DOC. These sources have formed since the maximum Holocene sea-level transgression and include wetland, soil, peat and palaeosols. It was found that there was not enough OM processing in the subsurface to significantly change the DOC character. The DOC character at this site therefore did not follow the regional chromatography model. All samples are more aromatic for their molecular weight in comparison to surface lakes and rivers and marine samples. It was also found that the physicochemical properties of the aquifer and in this case the anoxic, saturated conditions limited OM processes leading to enhanced preservation and storage of OM. The contradiction in DOM trends when compared to the redox chemistry of the groundwater, challenges our current understanding of groundwater DOC mobilisation and degradation in aquifers. A possible explanation for the observed constant character of the DOM with evolved inorganic redox chemistry is that all DOM fractions are being released from the OM sources at constant rates.

*In-situ* sources such as peats and palaeosols are the main contributor of OM to the groundwater DOC in this coastal system. However the destabilisation or changes in hydrology, whether by anthropogenic or natural processes could result in increased fluxes of carbon from deep within the peat column. Significantly, the results of this study show that understudied




anoxic coastal groundwater systems have the potential to export up to ten times more unreacted carbon to the surface than previously realised. These coastal groundwater sources are still unaccounted for in the global carbon budget and are likely to play more of a role in carbon transport in the future.

**7 Acknowledgements**

The authors would like to thank the NSW Office of Water for providing the funding for drilling and monitoring bore installation. Hamish Studholme, Sam McCulloch and Juan Carlos Castilla-Rho did the drilling and bore installation. NSW National Parks provided the scientific license and approval for the research to be conducted within the Tomaree National Park. Nur Syahiza Zainuddin was supported by a Malaysian government PhD scholarship. Thanks go to Ellen Howley for her help with the sampling. The authors would also like to thank various ANSTO personnel such as Robert Chisari and Kelly Farrawell for analysis.

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





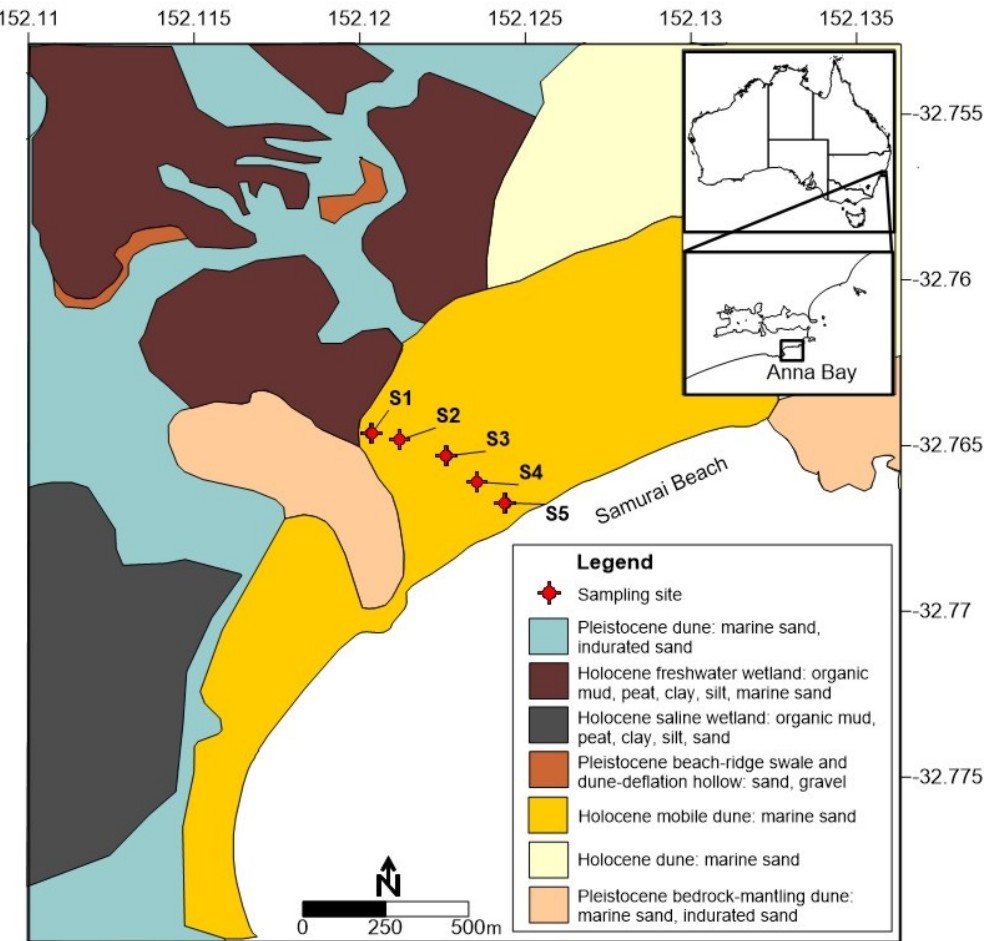

**Figure 1: Location of the study area and groundwater monitoring wells in relation to Coastal Quaternary Geology (Adapted from Hashimoto and Toedson, 2008).**





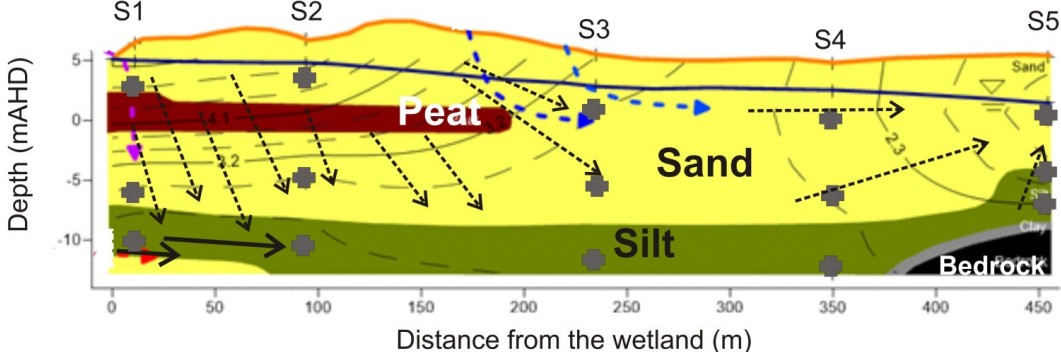

**Figure 2: Lithology and potentiometric contours in the aquifer in February 2014 (adapted from Maric, 2013 and Howley, 2014). Grey crosses are the location of screened intervals within the aquifer and arrows indicate groundwater flow direction.**

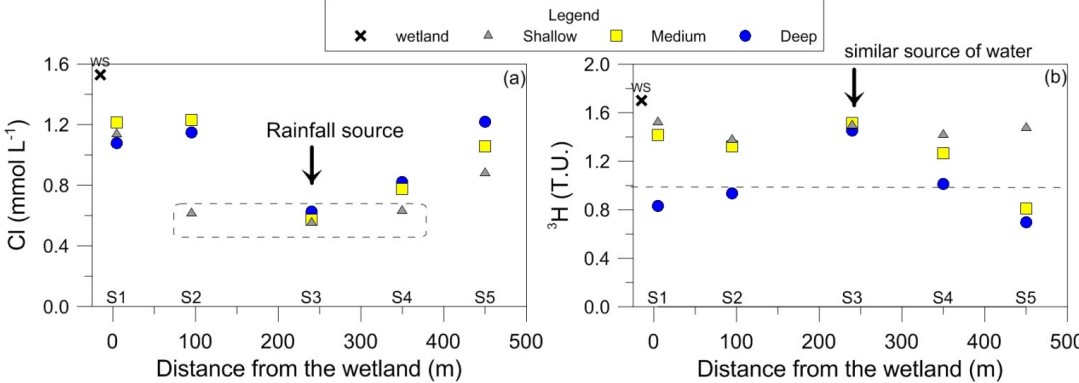

Figure 3: The relationship between (a) chloride concentration (mmol L[-1]) and the polygon represents rainfall values and (b) tritium (TU) content with distance from the wetland and the dotted line represents recently recharged groundwaters (i.e. 1-5 years old).





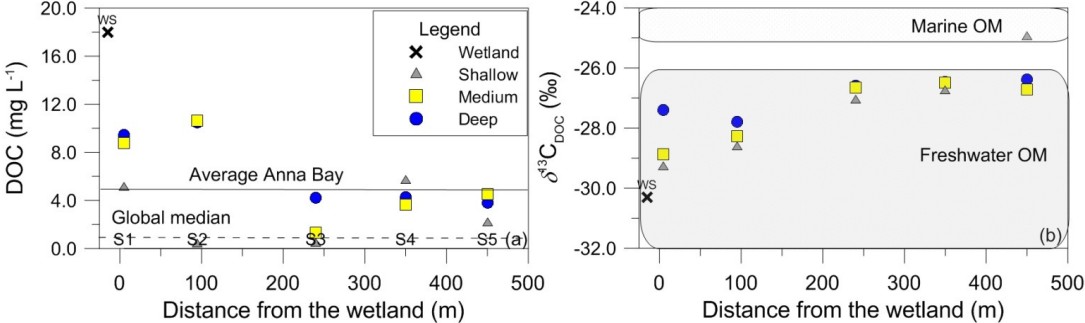

**Figure 4: The relationship between (a) DOC concentration compared to average DOC for Anna Bay (solid black line) and global median DOC concentration for groundwater (dotted line; McDonough et al., 2019) and (b) $\delta^{13}C_{DOC}$ values for groundwaters from Anna Bay with distance from the wetland and compared to the terrestrial and marine OM ranges from Lamb et al., (2006).**

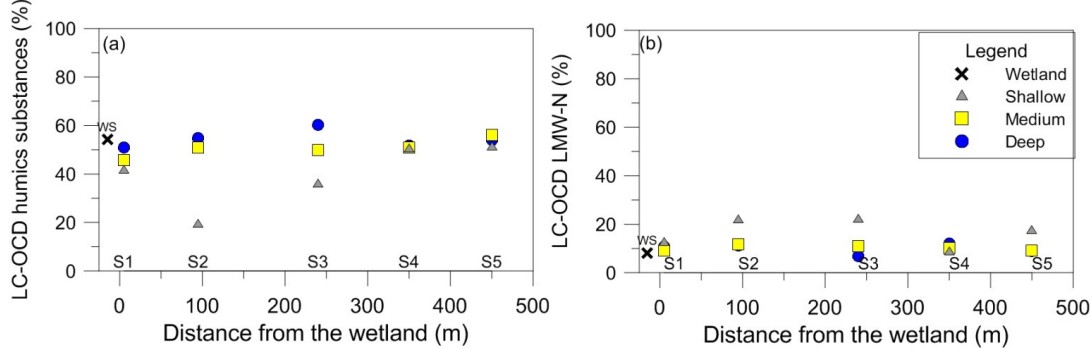

**Figure 5: The relationship between (a) LC-OCD results with the total percentage of humic substances and (b) LC-OCD total percentage of light molecular weight neutral with distance from the wetland.**





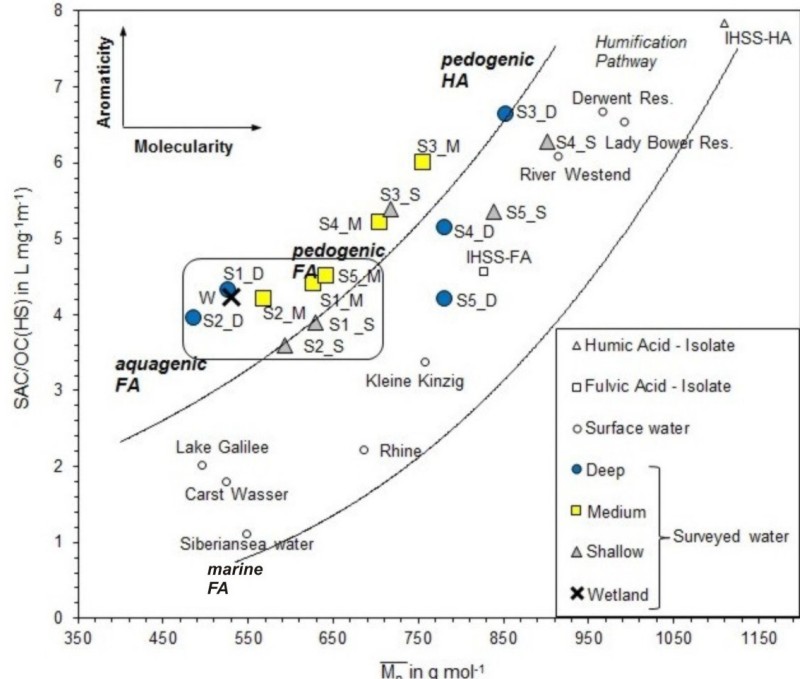

**Figure 6: The relationship between humic substances molecular weight ($M_n$) and humic substances aromaticity (SAC/OC = Spectral Absorption Coefficient/Organic Carbon). The polygon shows samples from Sites 1 and 2 plus the wetland sample, which have lower values than those from Sites 3-5. The curved lines show the area that samples would plot if the humic substances of aromaticity and molecular weight are derived from the humic substances standard and water samples from Huber et al., 2011. The figure also indicates the origin of humic substances samples plotting at the top interpreted as pedogenic in origin and the fulvic acids at the bottom are aquagenic origin. Note: humic acid and fulvic acid is isolated from humic substances standard of the IHSS from Suwannee River. Surface water samples with humic substances-aquagenic origin and humic substances-pedogenic origin are also indicated.**





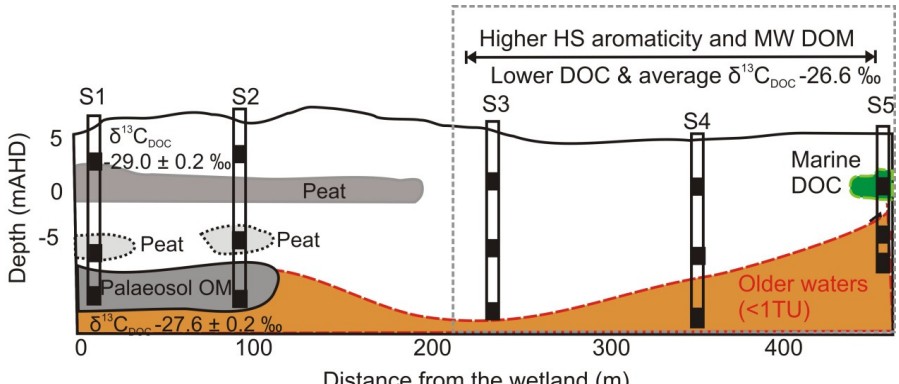

**Figure 7: Schematic of OM sources and processes influencing DOC within the coastal aquifer at Anna Bay based on concentration, character (isotopic and chromatographic), tritium and hydrochemical data (Table 1 and 2). Black open rectangles represent groundwater wells and filled rectangles are the location of the well screen ( ). Marine and freshwater DOM characterisation from**
5 **Lamb et al. (2006).**

**Table 1**: Water chemistry data for the site. Elevation of mid-point of the screen (m AHD), EC = electrical conductivity ($\mu$S cm$^{-1}$), DO = dissolved oxygen (mg L$^{-1}$) T = temperature (°C), Eh = redox potential (mV), and cations and anions (mg L$^{-1}$), n.a = not detected.

| ID | Elevation | Field pH | EC | DO | T | Eh | Ca | Fe | K | Mg | Mn | Na | Si | Sr | Cl | SO$_4$ | NO$_3$ | N-NH$_4$ |
|---|---|---|---|---|---|---|---|---|---|---|---|---|---|---|---|---|---|---|
| S1_S | 2.9 | 7.19 | 495 | 0.06 | 22.1 | -97.2 | 81.20 | 3.27 | 0.81 | 4.12 | 0.08 | 23.11 | 2.56 | 0.70 | 40.21 | n.a. | n.a. | 0.22 |
| S1_M | -6.1 | 6.1 | 337 | 0.04 | 19.6 | -14.5 | 38.60 | 1.40 | 0.97 | 7.96 | 0.02 | 24.05 | 2.11 | 0.35 | 43.03 | n.a. | n.a. | 0.14 |
| S1_D | -10.4 | 5.36 | 215 | 0.04 | 20.4 | 7.8 | 7.91 | 0.58 | 1.15 | 3.65 | 0.01 | 32.00 | 5.27 | 0.08 | 38.19 | n.a. | 0.18 | 0.80 |
| S2_S | 3.4 | 8.2 | 293 | 0.12 | 22.8 | -59.2 | 45.80 | 0.07 | 0.49 | 4.46 | 0.01 | 10.98 | 2.23 | 0.36 | 21.75 | 1.07 | n.a. | 0.00 |
| S2_M | -5.3 | 6.03 | 408 | 0.06 | 21.2 | -42.9 | 45.31 | 0.17 | 1.66 | 14.54 | 0.01 | 24.65 | 2.04 | 0.35 | 43.69 | n.a. | 0.16 | 0.40 |
| S2_D | -10.5 | 5.32 | 231 | 0.06 | 20.8 | -6.7 | 8.19 | 0.41 | 1.20 | 6.75 | 0.01 | 30.49 | 5.80 | 0.08 | 40.69 | n.a. | 0.28 | 0.90 |
| S3_S | 1.0 | 7.24 | 306 | 0.17 | 24.1 | -42.9 | 45.95 | 0.23 | 0.54 | 4.58 | 0.01 | 12.36 | 2.67 | 0.36 | 19.48 | 4.45 | n.a. | 0.03 |
| S3_M | -5.7 | 6.98 | 282 | 0.04 | 21.4 | -47.8 | 43.93 | 0.05 | 0.63 | 4.24 | 0.02 | 13.19 | 2.63 | 0.33 | 20.34 | 5.20 | n.a. | 0.08 |
| S3_D | -11.9 | 6.5 | 245 | 0.05 | 21.9 | -40.6 | 10.67 | 0.25 | 1.00 | 18.56 | 0.01 | 13.68 | 1.49 | 0.06 | 22.24 | 11.61 | n.a. | 0.81 |
| S4_S | 0.1 | 6.59 | 323 | 0.1 | 21.9 | -28.1 | 47.49 | 0.03 | 1.09 | 5.81 | 0.01 | 14.02 | 1.88 | 0.37 | 22.40 | 0.67 | n.a. | 0.57 |
| S4_M | -6.3 | 6.26 | 268 | 0.03 | 22 | -49 | 24.71 | 0.21 | 1.33 | 10.35 | 0.01 | 17.26 | 1.65 | 0.18 | 27.44 | 0.10 | n.a. | 0.84 |
| S4_D | -12.8 | 6.24 | 319 | 0.04 | 22.4 | 5.5 | 7.44 | 0.22 | 1.38 | 27.28 | 0.03 | 18.44 | 2.57 | 0.04 | 29.06 | n.a. | n.a. | 0.51 |
| S5_S | 0.1 | 6.69 | 365 | 0.06 | 21.8 | -62.5 | 51.22 | 0.71 | 1.10 | 5.96 | 0.00 | 19.13 | 2.43 | 0.42 | 31.16 | 10.18 | n.a. | 0.16 |
| S5_M | -4.6 | 6.1 | 316 | 0.04 | 22.1 | -46.5 | 7.89 | 0.31 | 2.25 | 21.38 | 0.03 | 26.61 | 3.39 | 0.06 | 37.46 | n.a. | 0.14 | 0.69 |




| ID | | | | | | | | | | | | | | | | | |
|---|---|---|---|---|---|---|---|---|---|---|---|---|---|---|---|---|---|
| S5_D | -7.4 | 6.16 | 348 | 0.1 | 21 | -58.4 | 8.10 | 0.59 | 2.89 | 22.39 | 0.04 | 34.13 | 3.72 | 0.07 | 43.15 | n.a. | 0.17 | 0.43 |
| WS | 5.3 | 5.48 | 200 | 1.01 | 21.7 | 87.1 | 8.00 | 0.43 | 2.18 | 2.74 | 0.01 | 28.01 | 1.23 | 0.07 | 54.23 | 3.93 | n.a. | 0.03 |

**Table 2**: Environmental isotope data for the site. DOC = Dissolved organic carbon, uncert = tritium uncertainty, QL = tritium quantification limit, CBE = charge balance error, $SI_{cc}$ = saturation index for calcite, PCO2 = partial pressure of carbon dioxide.

| ID | date | $\delta^{13}C_{DIC}$ | DOC | $\delta^{13}C_{DOC}$ | $^{14}C_{DIC}$ | $^{3}H$ | $^{3}H$ uncert | QL | CBE | DIC | $SI_{cc}$ | PCO₂ |
|---|---|---|---|---|---|---|---|---|---|---|---|---|
| | | (‰) | ppm | (‰) | pMC | TU | TU | TU | % | mmol/L | | atm |
| S1 _S | 18/02/2014 | -11.8 | 5.0 | -29.3 | 92.61 | 1.52 | 0.07 | 0.15 | 0.10 | 4.97 | 0.03 | 0.016 |
| S1_M | 18/02/2014 | -10.5 | 8.8 | -28.9 | 102.32 | 1.41 | 0.07 | 0.15 | 1.25 | 6.64 | -1.64 | 0.107 |
| S1_D | 18/02/2014 | -4.1 | 9.5 | -27.4 | 100.44 | 0.83 | 0.05 | 0.16 | 1.69 | 11.31 | -3.37 | 0.265 |
| S2_S | 19/02/2014 | -14.0 | 0.3 | -28.6 | 95.41 | 1.37 | 0.07 | 0.15 | -0.54 | 2.53 | 0.59 | 0.001 |
| S2_M | 18/02/2014 | -4.2 | 10.6 | -28.3 | 101.65 | 1.32 | 0.06 | 0.15 | 0.74 | 9.91 | -1.50 | 0.175 |
| S2_D | 18/02/2014 | -2.6 | 10.5 | -27.8 | 99.19 | 0.94 | 0.05 | 0.16 | 0.62 | 14.23 | -3.33 | 0.340 |
| S3_S | 19/02/2014 | -14.8 | 0.4 | -27.1 | 97.25 | 1.49 | 0.08 | 0.16 | 0.87 | 2.88 | -0.33 | 0.009 |
| S3_M | 19/02/2014 | -12.3 | 1.3 | -26.7 | 96.17 | 1.51 | 0.08 | 0.16 | 0.94 | 2.99 | -0.66 | 0.015 |
| S3_D | 19/02/2014 | -12.3 | 4.2 | -26.6 | 96.86 | 1.45 | 0.07 | 0.16 | 2.54 | 3.14 | -1.86 | 0.034 |
| S4_S | 20/02/2014 | -9.0 | 5.6 | -26.8 | 94.54 | 1.42 | 0.07 | 0.15 | -0.07 | 4.48 | -0.95 | 0.043 |
| S4_M | 20/02/2014 | -11.0 | 3.6 | -26.5 | 95.27 | 1.27 | 0.07 | 0.16 | 0.14 | 4.71 | -1.67 | 0.069 |
| S4_D | 20/02/2014 | -7.2 | 4.3 | -26.5 | 94.41 | 1.01 | 0.06 | 0.15 | 0.90 | 5.83 | -2.13 | 0.088 |
| S5_S | 21/02/2014 | -12.9 | 2.1 | -25.0 | | 1.47 | 0.07 | 0.16 | 0.89 | 4.13 | -0.83 | 0.034 |
| S5_M | 21/02/2014 | -9.1 | 4.5 | -26.7 | 97.78 | 0.81 | 0.05 | 0.16 | -0.34 | 6.49 | -2.28 | 0.111 |
| S5_D | 21/02/2014 | -9.4 | 3.8 | -26.4 | 96.86 | 0.69 | 0.04 | 0.16 | -0.47 | 6.66 | -2.19 | 0.105 |
| WS | 20/02/2014 | -23.6 | 18.0 | -30.3 | | 1.70 | 0.08 | 0.15 | 4.40 | 1.56 | -3.96 | 0.037 |

