# Peer review of "Isotopic and chromatographic fingerprinting of the sources of dissolved organic carbon in a shallow coastal aquifer"

_Hydrology and Earth System Sciences, 2018_

## Referee Comment (RC1) · Hannelore Waska (Referee) · 16 Oct 2019

Overall: Please proof-read the MS carefully, there seem to be a few small typos/grammatical errors.

Data analysis: The dataset is well-suited for a multivariate data analysis to decipher GW sources (see RDA as example in cited Coutourier et al. 2016). I recommend the inclusion of a multivariate analysis (RDA or PCA).

Methods, p4, l15 "Dissolved". Also, I think the company name is Waterra.

p5, l10 ff: Why was GW age (years) not calculated? What is the merit of using TU?

[Figure]

All methods: Please include details for 14C analysis. Why was 14C of DOC not measured?

Results, p5, l25: Would the authors expect seawater infiltration, due on tidal inundation and/or storm floods, at S5? Or is the GW pressure so high that it immediately dilutes any seawater influence?

All results and following discussions: Please use either present or past tense continuously throughout. Results contain interpretations (e.g. indications of marine carbonate dissolution...ion exchange processes...methanogenesis...) which may better fit in the Discussion section.

p7, l1: "The average DOC concentration (...) is high" compared to what? It is not high considering the conditions (anoxic, advective flow, peat hydrolysis in the aquifer).

Section 4.1., first two paragraphs: Please refrain from switching between past and present tenses.

Discussion, p9, l5-8: Please provide additional literature which supports your claim of a global occurrence. I have added some examples to the reference list below.

Overall Discussion: It seems that 14C-DIC is not included in the discussion of the results. Why? How can it help in interpreting GW sources?

Conclusion, p12, l1ff: Please explain how the estimate of an "order of magnitude higher" is achieved.

p13, l1: Please explain how the estimate of an "export up to ten times" is achieved.

Figure 2: Perhaps there is a way to improve the quality of the figure (some features appear to be blurred). What do the blue, pink, and red arrows mean in contrast to the black ones?

References

Coutourier M, Nozais C, Chaillou G, 2016, Microtidal subterranean estuaries as a source of fresh terrestrial dissolved organic matter to the coastal ocean. Marine Chemistry 186, 46-57.

Goñi MA, Gardner IR, 2003, Seasonal dynamics in dissolved organic carbon concentrations in a coastal water-table aquifer at the forest-marsh interface. Aquatic Geochemistry 9, 209-232.

Seibert SL, Holt T, Reckhardt A, Ahrens J, Beck M, Pollmann T, Giani L, Waska H, Böttcher ME, Greskowiak J, Massmann G, 2018, Hydrochemical evolution of a freshwater lens below a barrier island (Spiekeroog, Germany): The role of carbonate mineral reactions, cation exchange and redox processes. Applied Geochemistry 92, 196-208.

Streif H, 2004, Sedimentary record of Pleistocene and Holocene marine inundations along the North Sea coast of Lower Saxony, Germany. Quarternary International, 112, P3-28.
* * *

---

## Referee Comment (RC2) · Anonymous Referee #2 · 20 Dec 2019

General Comments:

In this paper, Meredith et al. describe the age of groundwater and composition of DOM in a coastal anoxic aquifer in coastal New South Wales. They propose several DOM sources present within the aquifer, and suggest that anoxic aquifers can contain substantially more DOC than other aquifers and that further research is needed to understand the stabilization of and potential fates of the DOM contained in anoxic aquifers globally.

The paper is short and easy to read, and provides important data on anoxic coastal aquifer chemistry to the scientific literature.

[Figure]

Overall, the similarity of DOM composition in the aquifer to the DOM in the wetland seems to need more addressing, particularly given the claim the paper makes that the wetland is not the source of DOM for the aquifer. It would strengthen the paper to further discuss possible reasons (including the possible lack of compositional resolution when using only LC-OCD to assess DOM composition) why the DOM composition is the same, yet the paper claims the sources are be different.

Specific Comments:

My main scientific concern about the paper is the claim that DOM is not being processed in these systems. On page 7 line 24-page 8 line 1, a claim is made that DOM would be expected to decrease in DOM aromaticity if it were being biodegraded. I'm not sure this statement holds true for all DOM systems. If DOM were being biodegraded, it could likely become more aromatic along a flowpath due to preferential microbial consumption of more aliphatic, biolabile molecules. I think this statement needs to be backed up by some citations or expanded to explain the degradation pathways being discussed, since higher aromaticity to me suggests preferential microbial processing of biolabile aliphatic fractions is occurring as the DOM moves through the groundwater.

Similarly, I do not understand the reasoning presented on page 10 line 6-10. The paper says that the d13C value of DOC should get lighter due to microbial processing, and that yes, in fact the groundwater DOC d13C values are 1 ppm lighter than those in the wetland, yet the paper seems to be saying that this goes against the idea of microbial fractionation. Doesn't that show that yes, the d13C-DOC values are becoming lighter than the original C3 vegetation wetland source, which would support biological processing as a mechanism at work here? I may have missed a piece of logic, but it would strengthen the point of the paper if this were clarified. Further, I do not follow the line of reasoning that claims the modern wetland is not a major source of DOM for these groundwaters on page 10 line 15-16. It appears that DOM composition and d13C are consistent with a wetland source. If the water chemistry is different, it is probable that the DOM source and water source are decoupled, which is interesting. But I do not

think the claim that the modern wetland is not the source of DOM to the groundwater is justified based on the explanation of the data provided. Perhaps a more detailed explanation of why the isotopes lead the authors to this conclusion would strengthen the argument. Other specific comments:

Section 4 is a Results and Discussion section, not a pure Results section, and should be labeled accordingly or take some of the discussion out and put it later in the discussion section—I find it a little confusing cause important discussion material is listed here, but then there is a separate discussion section

Page 7 line 22-24: Was the intention to say that groundwater from sites 3-5 had higher aromaticity than other groundwater/shallower groundwater samples, or higher aromaticity than other surface water samples in other studies? If the former (which is how I initially read this), this seems to contradict the statement in the discussion that the groundwater DOM composition is constant in this study.

Page 11 lines 19-22: It might strengthen the claim that this carbon may be important for the regional carbon budget to explicitly relate this anoxic, preserved DOM to freshly produced DOM, and to explicitly state how it may impact the regional carbon budget. The claim is made in the conclusion and the abstract that anoxic coastal groundwater systems have the potential to export up to ten times more unreacted carbon than thought, but the calculation or logic to support this is never shown. It would strengthen the claim to demonstrate why that claim is being made, and how the math adds up if the carbon concentrations in the groundwater are only five times higher than those in other aquifers. In general LC-OCD may give too low a resolution to reveal DOM compositional differences between sources and groundwater—perhaps a discussion of other areas where LC-OCD on its own has been able to tease out compositional differences could strengthen the expectation that if there were compositional differences here, LC-OCD could assess them. Twice it's mentioned that the aromaticity for the humic substances is higher than for lakes or rivers (once in 5.1.2, once in 41.). Especially in the results, it would be useful to see the values from the Huber et al 2011 study and

this study to show how they compare. Technical Corrections:

Page 4 line 9: "may be" not "maybe"

Page 4 line 17: "Dissolved" not "Dissolve"

Page 9 line 5: Organic should not be capitalized

Figure 3: "the polygon" is unclear—perhaps call it the "dashed rectangle" Figure 6: "Siberian Sea" not "Siberiansea"
* * *

---

## Author Comment (AC1) · 30 Dec 2019

RC1 - Hannelore Waska (Referee) hannelore.waska@uol.de

Overall: Please proof-read the MS carefully, there seem to be a few small typos/ grammatical errors. Author's response: The MS has been proof read. Small typos, etc have been found throughout and adjusted. The MS has been corrected to read in the past tense.

Data analysis: The dataset is well-suited for a multivariate data analysis to decipher

GW sources (see RDA as example in cited Coutourier et al. 2016). I recommend the inclusion of a multivariate analysis (RDA or PCA). Author's response: The authors agree and have included a PCA. This analysis will be included in the final MS. The methods for PCA were added to page 6 line 5 – "Principal Component Analysis (PCA) was performed in base R. The data was centred to the mean of the variable and then scaled using the variable standard deviations in R using the prcomp function: https://stat.ethz.ch/R-manual/R-devel/library/stats/html/prcomp.html." The following will be added to the results section page 7 ln 21- "Principal component analysis (Fig. 6) using water quality, isotopes and DOC variables including LC-OCD fractions (%), ðİŻ£13CDOC (‰, ðİŻ£13CDIC (‰, 3H, pH, Na, pCO2, NO3, Cl, Ca, Sr, DO, SO4, NH4, Ca and DOC concentration (mg C / L) confirms the presence of different groundwater sources." The variables contributing to PC1 (in order of importance) are pH, Na, pCO2, NO3, Cl, Ca, DOC, 3H, Sr and humics. The variables contributing to PC2 in order of importance are ðİŻ£13CDOC, DO, biopolymers, ðİŻ£13CDIC, HS aromaticity, HS mol weight, DOC, Cl and 3H. PC1 mainly explains the variations we see with sample depth. Samples S1_D, S2_S, S2_D and S3_S are the samples most strongly influencing PC1. The deep samples that are likely to have originated from a deeper regional source of water (S1_D and S2_D) are influencing the right hand side of the PCA with high pCO2, NO3, humics, Na, and Cl. These samples are also characterised by low Ca, 3H, Sr and pH. The shallow samples (S2_S and S3_S) are influencing the left hand side of the PCA with low pCO2, NO3, humics, Na and Cl, and have high Ca, 3H, Sr and pH suggestive of rainfall recharge waters. This analysis further highlights the distinct wetland sample that is not related to the other samples (Fig 6) and is heavily influencing PC2 with low ðİŻ£13CDOC values, high DO, high biopolymers, low HS aromaticity, low HS molecular weight and high DOC concentration."

New Fig 6 added to text Methods, p4, l15 "Dissolved". Author's response: Corrected

Also, I think the company name is Waterra. Author's response: Corrected p5, l10 ff: Why was GW age (years) not calculated? What is the merit of using TU? Author's

response: The raw 3H measurement value was used as a guide for recent recharge. Rainfall for the regions is expected to be 1.7TU as stated page 6, line 16. If the sample contained values close to rainfall then it can be interpreted that the groundwater contains recent rainfall. The groundwater age was not calculated because there was no rainfall or time-series analysis collected for the system. Groundwater age calculations are non-unique if calculated on a single sample event. Generally when calculating a groundwater age using 3H it is advised to use a lumped parameter model such as what we did in our time-series study of Rottnest island lens using 3H as an age tracer of groundwater age (Bryan et al., 2019). Therefore the authors use 3H as a tracer of rainfall recharge and do not calculate a groundwater age. The following sentence will be added to the final MS "Tritium activities were used as an indication of groundwater recharge occurrence by rainfall and groundwater ages were not calculated due to a lack of time series data collected for this study. Bryan et a. (2019) shows the importance of collecting 3H data and then calculating a groundwater age using a lumped parameter model in a shallow unconfined aquifer."

All methods: Please include details for 14C analysis. Why was 14C of DOC not measured? Author's response: Please note all reference to 14CDIC will be removed from the paper. This tracer indicates that the groundwaters are all modern similar to 3H and 14CDIC is not useful in such a young groundwater environment. The 14C of DOC was measured in a later study by McDonough et al., 2020 accepted in Geochimica et Cosmochimica Acta. Reference to this paper will be made in the discussion section. The 14CDOC results for sites 1 and 2 were:

S1_M S1_D S2_D 14CDOC (pMC) 100.86 88.69 87.94 These results suggest an older peat source at depth compared to the shallow samples. Please note that the results for these samples were processed and submitted in mid-2019, well after this MS was submitted for review in Dec 2018 to HESS.

The reason 14CDOC was not analysed for this study is that it was conducted in 2010, this radioactive isotope was not available for measurement at ANSTOs AMS facility
during this time. We later underwent methods development to add this isotope to our methods development.

Results, p5, l25: Would the authors expect seawater infiltration, due on tidal inundation and/or storm floods, at S5? Or is the GW pressure so high that it immediately dilutes any seawater influence? Author's response: The water chemistry and hydraulic head data suggests S5 is fresh water. The following sentence was added to page 6 ln 5 to clarify this point "and there was no evidence of seawater infiltration after storm events based on the hydrochemical data."

All results and following discussions: Please use either present or past tense continuously throughout. Author's response: The MS has been checked for tense and corrected to past tense.

Results contain interpretations (e.g. indications of marine carbonate dissolution: : :ion exchange processes...methanogenesis: : :) which may better fit in the Discussion section. Author's response: Agreed. This section has been removed from page 7 lines 1-7 because it is discussed in detail later in the discussion section. The following sentence was added to page 6 ln 20 to describe the results rather than provide discussion "The DIC values also showed various sources and processes influencing inorganic carbon."

p7, l1: "The average DOC concentration (: : :) is high" compared to what? It is not high considering the conditions (anoxic, advective flow, peat hydrolysis in the aquifer). Author's response: The following sentence has been added "compared to the $\sim$1 mg L-1 for the global median DOC concentration in groundwater (McDonough et al., 2019)."

Section 4.1., first two paragraphs: Please refrain from switching between past and present tenses. Author's response: Agreed. This has been corrected throughout the MS.

Discussion, p9, l5-8: Please provide additional literature which supports your claim of a global occurrence. I have added some examples to the reference list below. Author's

response: McDonough et al. (2019) presents the largest global dataset of 7,849 published and unpublished groundwater DOC concentrations. They calculate the global median DOC concentration for groundwater based on this dataset. The papers provided have been reviewed by the authors and considered for the corrections.

McDonough, L., Santos, I., Andersen, M., O'Carroll, D., Rutlidge, H., Meredith, K. and Oudone, P.: "Changes in Global Groundwater Organic Carbon Driven by Climate Change and Urbanization." EarthArXiv. November 21. doi:10.31223/osf.io/vmaku, 2018.

Overall Discussion: It seems that 14C-DIC is not included in the discussion of the results. Why? How can it help in interpreting GW sources? Author's response: The 14CDIC has been removed from this MS. The results all show a modern source of DIC, similar to the 3H results. Please see the description above for a detailed response.

Conclusion, p12, l1ff: Please explain how the estimate of an "order of magnitude higher" is achieved. Author's response: The concentrations found in this coastal system contain up to 10 mg/L of DOC whereas the global medium is 1 mg/L, hence the description of an order or magnitude. The following has been added to the text to describe this further Page 13, ln 20 "The average groundwater DOC concentration for this study was five times higher (5 mg L-1) than the global median DOC concentration for groundwaters. The concentration of DOC doubled with depth, reaching 10 mg L-1 but the DOM chromatographic character did not change significantly with depth or along the groundwater flow path but the carbon isotopic composition did change."

p13, l1: Please explain how the estimate of an "export up to ten times" is achieved. Author's response: The concentrations found in this coastal system contain up to 10 mg/L of DOC whereas the global medium is 1 mg/L, see above comment.

Figure 2: Perhaps there is a way to improve the quality of the figure (some features appear to be blurred). What do the blue, pink, and red arrows mean in contrast to the black ones? Agreed. The Figure has been updated and the various coloured arrows

have not been removed for clarity.

[Figure]

**Fig. 1.** Fig 6

[Figure]

**Fig. 2.** Fig 2 new

---

## Author Comment (AC2) · 30 Dec 2019

RC2 - Anonymous Referee #2 General Comments: In this paper, Meredith et al. describe the age of groundwater and composition of DOM in a coastal anoxic aquifer in coastal New South Wales. They propose several DOM sources present within the aquifer, and suggest that anoxic aquifers can contain substantially more DOC than other aquifers and that further research is needed to understand the stabilization of and potential fates of the DOM contained in anoxic aquifers globally. The paper is short and easy to read, and provides important data on anoxic coastal aquifer chemistry to the scientific literature. Overall, the similarity of

[Figure]

DOM composition in the aquifer to the DOM in the wetland seems to need more addressing, particularly given the claim the paper makes that the wetland is not the source of DOM for the aquifer. It would strengthen the paper to further discuss possible reasons (including the possible lack of compositional resolution when using only LC-OCD to assess DOM composition) why the DOM composition is the same, yet the paper claims the sources are be different. Author's response: The limits to the compositional resolution of LC-OCD could explain the small changes in chromatographic character. High resolution mass spectrometric techniques such as FT-ICR-MS have been used in this study site to further probe changes in groundwater DOC character along the transect. Recent research has demonstrated its utility in identifying groundwater DOC compositional changes after recharge at the site (McDonough et al 2020). The isotopic differences and the newly added PCA analysis clearly show that the sources of DOC in the aquifer are different. These new lines of evidence will be discussed in the final MS. A paper by (McDonough et al 2020) accepted in GCA also shows that the DOM has different ages shallow 100 pmc and deeper source 80 pmc. Reference to these recent papers will be made within the final MS discussion. These papers are newly published and were worked completed while this submission was under review with HESS.

The following analysis will be added to the results section. "Principal component analysis (Fig. 6) using water quality, isotopes and DOC variables including LC-OCD fractions (%), ðİŻ£13CDOC (‰, ðİŻ£13CDIC (‰, 3H, pH, Na, pCO2, NO3, Cl, Ca, Sr, DO, SO4, NH4, Ca and DOC concentration (mg C / L) confirms the presence of different groundwater sources." The variables contributing to PC1 (in order of importance) are pH, Na, pCO2, NO3, Cl, Ca, DOC, 3H, Sr and humics. The variables contributing to PC2 in order of importance are ðİŻ£13CDOC, DO, biopolymers, ðİŻ£13CDIC, HS aromaticity, HS mol weight, DOC, Cl and 3H. PC1 mainly explains the variations we see with sample depth. Samples S1_D, S2_S, S2_D and S3_S are the samples most strongly influencing PC1. The deep samples that are likely to have originated from a deeper regional source of water (S1_D and S2_D) are influencing the right hand side of the PCA with high pCO2, NO3, humics, Na, and Cl. These samples are also characterised

by low Ca, 3H, Sr and pH. The shallow samples (S2_S and S3_S) are influencing the left hand side of the PCA with low pCO2, NO3, humics, Na and Cl, and have high Ca, 3H, Sr and pH suggestive of rainfall recharge waters. This analysis further highlights the distinct wetland sample that is not related to the other samples (Fig 6) and is heavily influencing PC2 with low ðİŻ£13CDOC values, high DO, high biopolymers, low HS aromaticity, low HS molecular weight and high DOC concentration." The reference to the new work completed: McDonough, L.K., O'Carroll, D.M., Meredith, K., Andersen, M.S., Brügger, C., Huang, H., Rutlidge, H., Behnke, M.I., Spencer, R.G.M., McKenna, A., Marjo, C.E., Oudone, P. and Baker, A. (2020) Changes in groundwater dissolved organic matter character in a coastal sand aquifer due to rainfall recharge. Water Research 169, 115201.

Specific Comments: My main scientific concern about the paper is the claim that DOM is not being processed in these systems. On page 7 line 24-page 8 line 1, a claim is made that DOM would be expected to decrease in DOM aromaticity if it were being biodegraded. I'm not sure this statement holds true for all DOM systems. If DOM were being biodegraded, it could likely become more aromatic along a flowpath due to preferential microbial consumption of more aliphatic, biolabile molecules. I think this statement needs to be backed up by some citations or expanded to explain the degradation pathways being discussed, since higher aromaticity to me suggests preferential microbial processing of biolabile aliphatic fractions is occurring as the DOM moves through the groundwater. Author's response: Agreed. This sentence will be changed to read: "DOC would be expected to have changed in aromaticity and molecular weight along a flow line, the opposite to what is observed here if the samples formed a degradation pathway". Page 8 lines 10-12

The MS has been reviewed extensively and all reference to the DOM character being constant is removed from the MS. It was not the aim of the MS to suggest the DOM is constant. The DOM character varies in the aquifer, but does not vary along hypothesised or expected flowpaths. This just further highlights that there are distinctive

sources of DOM in the aquifer and not one wetland source recharging the aquifer and biodegrading along a 500 m transect.

To address the comment that microbial degradation might be expected to break down the aliphatic fraction. We will reference the importance of sorption and biodegradation in the corrected MS. We could cite Chapelle et al (Hydrogeol. J) as this paper shows changes in optical clarity along a flowpath that are substantial. We also add the experimental work of Oudone et al., 2019, which confirms that sorption predominantly affects the aromatic, HS fraction). For biodegradation, we cite Shen, Chappelle et al 2014 which shows that only a small proportion of groundwater DOC is bioavailable. The new reference of Oudone, P., Rutlidge, H., Andersen, M.S., O'Carroll, D., Cheong, S., Meredith, K., McDonough, L., Marjo, C. and Baker, A. (2019) Characterisation and controls on mineral-sorbed organic matter from a variety of groundwater environments. EarthArXiv https://eartharxiv.org/ue86w/.

We have rewritten the following sentence to highlight . . .. "from S3, S4 and S5_S also had higher humic substances aromaticity and humic substances molecular weight than sites 1 and 2." Page 8 line 10

Page 8, ln 20 was added to read "Furthermore, despite observed differences in DOM characteristics (e.g. higher humic substances aromaticity and molecular weight in S3, S4 and S5S), DOM character does not change significantly along the groundwater flow path, contrary to what was found in other studies where biodegradation, sorption, desorption and biosynthesis controlled DOM (Chappelle et al., 2012; Shen et al., 2015)."

Similarly, I do not understand the reasoning presented on page 10 line 6-10. The paper says that the d13C value of DOC should get lighter due to microbial processing, and that yes, in fact the groundwater DOC d13C values are 1 ppm lighter than those in the wetland, yet the paper seems to be saying that this goes against the idea of microbial fractionation. Doesn't that show that yes, the d13C-DOC values are becoming lighter than the original C3 vegetation wetland source, which would support biological

processing as a mechanism at work here? I may have missed a piece of logic, but it would strengthen the point of the paper if this were clarified. Author's response: The groundwater 13CDOC are heavier than the wetland source. The wetland source at the surface is likely to have undergone fractionation due to microbes. Reference to this are made clearer in the MS.

Page 10, ln 23-25 will be added "This suggests that if the wetland was the source of OM then the carbon isotopes were fractionated after recharge and become heavier than the wetland OM". "Alternatively, the difference in isotopic values can be explained by suggesting the DOM in the groundwater system has a different source to the wetland and there is limited interaction of the surface and groundwater."

Further, I do not follow the line of reasoning that claims the modern wetland is not a major source of DOM for these groundwaters on page 10 line 15-16. It appears that DOM composition and d13C are consistent with a wetland source. If the water chemistry is different, it is probable that the DOM source and water source are decoupled, which is interesting. But I do not think the claim that the modern wetland is not the source of DOM to the groundwater is justified based on the explanation of the data provided. Perhaps a more detailed explanation of why the isotopes lead the authors to this conclusion would strengthen the argument. Author's response: The isotopes do not suggest consistent sources of DOM. Water sources have become decoupled from the different to the DOM sources. Further studies completed by McDonough et al., 2020 will be included in the discussion with FT-ICR-MS evidence. We have also found in further time-series studies of the site that desorption of SOM after recharge is occurring. These new findings will be incorporated into the final discussion to strengthen the discussion on sources of DOM. It must be noted these studies have been completed while this MS has been under review.

Other specific comments: Section 4 is a Results and Discussion section, not a pure Results section, and should be labeled accordingly or take some of the discussion out and put it later in the discussion sectionâĚŸAĚĞ TI find it a little confusing cause

important discussion material is listed here, but then there is a separate discussion section Author's response: Agreed. This section has been removed from page 7 lines 1-7 because it is discussed in detail later in the discussion section. This point was also addressed by R1. The following sentence was added to page 6 ln 20 to describe the results rather than provide discussion "The DIC values also showed various sources and processes influencing inorganic carbon."

Page 7 line 22-24: Was the intention to say that groundwater from sites 3-5 had higher aromaticity than other groundwater/shallower groundwater samples, or higher aromaticity than other surface water samples in other studies? If the former (which is how I initially read this), this seems to contradict the statement in the discussion that the groundwater DOM composition is constant in this study. Author's response: It was not the intention of this study to suggest the DOM is constant as mentioned above, the DOM does not vary along hypothesised or expected flowpaths.

Page 11 lines 19-22: It might strengthen the claim that this carbon may be important for the regional carbon budget to explicitly relate this anoxic, preserved DOM to freshly produced DOM, and to explicitly state how it may impact the regional carbon budget. The claim is made in the conclusion and the abstract that anoxic coastal groundwater systems have the potential to export up to ten times more unreacted carbon than thought, but the calculation or logic to support this is never shown. It would strengthen the claim to demonstrate why that claim is being made, and how the math adds up if the carbon concentrations in the groundwater are only five times higher than those in other aquifers. In general LC-OCD may give too low a resolution to reveal DOM compositional differences between sources and groundwaterâËŸA ËĞ Tperhaps a discussion of other areas where LC-OCD on its own has been able to tease out compositional differences could strengthen the expectation that if there were compositional differences here, LC-OCD could assess them. Twice it's mentioned that the aromaticity for the humic substances is higher than for lakes or rivers (once in 5.1.2, once in 41.). Especially in the results, it would be useful to see the values from the Huber et al 2011 study and

this study to show how they compare. Author's response: The groundwater results are shown with respect to Huber's study please refer to Fig 7. More emphasis on this figure will be added to the final MS.

The new addition from Oudone et al., (2019) experimental work, clearly shows how LC-OCD can elucidate sorption processes. Rutlidge et al (2015) GCA shows how it could elucidate DOM sources and transformation during recharge using an artificial rainfall event. McDonough use LC-OCD and FT-ICRMS to understand changes in DOM character after rainfall recharge. A summary of these works will be included in the final discussion.

Technical Corrections: Page 4 line 9: "may be" not "maybe" Author's response: corrected Page 4 line 17: "Dissolved" not "Dissolve" Author's response: corrected Page 9 line 5: Organic should not be capitalized Author's response: corrected Figure 3: "the polygon" is unclearâËŸAËĞ Tperhaps call it the "dashed rectangle" Author's response: corrected Figure 6:"Siberian Sea" not "Siberiansea" Author's response: corrected

---

## Author Response (AR2)

**RC1 -**
**Hannelore Waska (Referee)**
hannelore.waska@uol.de

Overall: Please proof-read the MS carefully, there seem to be a few small typos/
grammatical errors.
Author's response: The MS has been proof read. Small typos, etc were found throughout and adjusted.
The MS has been corrected to read in the past tense.

Data analysis: The dataset is well-suited for a multivariate data analysis to decipher
GW sources (see RDA as example in cited Coutourier et al. 2016). I recommend the
inclusion of a multivariate analysis (RDA or PCA).
Author's response: The authors agree and have included a PCA analysis.

The methods for PCA is added to page 6 line 1 –
"Principal Component Analysis (PCA) was performed in base R. Parameters which were consistently above their limit of detection were investigated by PCA. The data was centred to the mean of the variable and then scaled using the variable standard deviations in R using the prcomp function: https://stat.ethz.ch/R-manual/R-devel/library/stats/html/prcomp.html."

20 The following will be added to the results section page 7 starting line 8 -
"Principal component analysis (Fig. 6) using water quality parameters, isotopes and DOC variables including LC-OCD fractions (%), $\delta^{13}C_{DOC}$ (‰), $\delta^{13}C_{DIC}$ (‰), $^3H$, pH, Na, $pCO_2$, Cl, Ca, Sr, DO, $SO_4$, $NH_4$, Ca and DOC concentration confirmed the presence of different groundwater sources. The variables contributing to PC1 (in order of importance) were pH, Na, $pCO_2$, Cl, DOC, Ca, humics,

25 LMW-N, Sr and $^3H$. The variables contributing to PC2 in order of importance are $\delta^{13}C_{DOC}$, DO, biopolymers, HS aromaticity, $\delta^{13}C_{DIC}$, HS molecular weight, Cl, DOC and hydrophobic DOM. PC1 mainly explained the variations with sample depth in the aquifer. Samples S1_D, S2_S, S2_D and S3_S were the samples that most strongly influenced PC1. The deep samples likely to have originated from a deeper regional source of water (S1_D and S2_D) were influencing the right hand side of the PCA with

30 high $pCO_2$, humics substances, Na and Cl. These samples were also characterised by low Ca, $^3H$ and Sr. The shallow samples (S1_S, S2_S and S3_S) were influencing the left hand side of the PCA with low $pCO_2$, humics, Na and Cl, and have high Ca, $^3H$, and Sr suggestive of rainfall recharge waters. This analysis further highlights the wetland sample that is not related to the other samples (Fig 6) and is heavily influencing PC2 with low $\delta^{13}C_{DOC}$ values, high DO, high biopolymers, low HS aromaticity, low

35 HS molecular weight and high DOC concentration."

[Figure]

New Fig 6 added to text
Methods, p4, l15 "Dissolved".
Author's response: Corrected

Also, I think the company name is Waterra.
Author's response: Corrected

p5, l10 ff: Why was GW age (years) not calculated? What is the merit of using TU?

Author's response: The raw 3H measurement was used as a guide for recent recharge. Rainfall for the
10  regions is expected to be 1.7 TU as stated in page 6, line 16. If the sample contained values close to
rainfall then it can be interpreted that the groundwater contains recent rainfall. The groundwater age
was not calculated because there was no rainfall or time-series analysis collected for the system.
Groundwater age calculations are non-unique if calculated on a single sample event. Generally when
calculating a groundwater age using 3H it is advised to use a lumped parameter model such as what we
15  did in our time-series study of Rottnest island lens using 3H as an age tracer of groundwater age (Bryan
et al., 2020). Therefore the authors use 3H as a tracer of rainfall recharge and do not calculate a
groundwater age.
The following sentence was added to page 5 line 14
"Tritium activities were used as an indication of groundwater recharge occurrence by rainfall and
20  groundwater ages were not calculated due to a lack of time series data collected for this study. Bryan et

a. (2020) shows the importance of collecting $^{3}$H time-series data and then calculating a groundwater age using a lumped parameter model in a shallow unconfined aquifer."

All methods: Please include details for 14C analysis. Why was 14C of DOC not measured?

Author's response: Please note all reference to $^{14}$C$_{DIC}$ was removed from the paper. This tracer indicated the groundwaters are modern and 14CDIC is not useful for calculating groundwater residence times in such a young groundwater environment. The 14C of DOC was measured in a later study by McDonough et al., (2020), on a different dataset. Reference to this paper is made in the discussion section. Page 10, line 11. The 14CDOC results for sites 1 and 2 were:

|  | S1_M | S1_D | S2_D |
|---|---|---|---|
| $^{14}$C$_{DOC}$ (pMC) | 100.86 | 88.69 | 87.94 |

These results suggest an older peat source at depth compared to the shallow samples. Please note that the results for these samples were processed and submitted in mid-2019 to a different journal, well after this MS was submitted for review in Dec 2018 to HESS.

The reason 14CDOC was not analysed for this study is that the field samples were collected and analysed in 2014, this method was not available for measurement at ANSTOs AMS facility at that time.

Results, p5, l25: Would the authors expect seawater infiltration, due on tidal inundation and/or storm floods, at S5? Or is the GW pressure so high that it immediately dilutes any seawater influence?

Author's response: The water chemistry data suggests at S5, the most seaward monitoring bore, is fresh water. The following sentence was added to page 6, line 7 to clarify this point

"and there was no evidence of seawater infiltration after storm events based on the hydrochemical data."

All results and following discussions: Please use either present or past tense continuously throughout.

Author's response: The MS was checked for tense and corrected to past tense.

Results contain interpretations (e.g. indications of marine carbonate dissolution: : :ion exchange processes...methanogenesis: : :) which may better fit in the Discussion section.

Author's response:

Agreed. This section was removed from page 7 lines 1-7 because it is discussed in detail later in the discussion section.

p7, l1: "The average DOC concentration (: : :) is high" compared to what? It is not high considering the conditions (anoxic, advective flow, peat hydrolysis in the aquifer).

Author's response:

The following sentence was added page 6 line 20
"compared to the ~1 mg L$^{-1}$ for the global median DOC concentration in groundwater (McDonough et al., 2019)."

Section 4.1., first two paragraphs: Please refrain from switching between past and present tenses.
Author's response:
Agreed. This has been corrected throughout the MS.

Discussion, p9, l5-8: Please provide additional literature which supports your claim of a global occurrence. I have added some examples to the reference list below.
Author's response:
McDonough et al. (2019) presents the largest global dataset of 7,849 published and unpublished groundwater DOC concentrations. They calculate the global median DOC concentration for groundwater based on this dataset.

Overall Discussion: It seems that 14C-DIC is not included in the discussion of the results. Why? How can it help in interpreting GW sources?
Author's response:
The 14CDIC was removed from this MS. The results show a modern source of DIC, similar to the 3H. Please see the description above for a detailed response.

Conclusion, p12, l1ff: Please explain how the estimate of an "order of magnitude higher" is achieved.
Author's response:
The concentrations found in this coastal system contain up to 10 mg/L of DOC whereas the global medium is 1 mg/L, hence the description of an order or magnitude. The following has been added to the text to clarify:
page 13, line 3-4 "The average groundwater DOC concentration for this study was five times higher (5 mg L$^{-1}$) than the global median DOC concentration for groundwaters. The concentration of DOC doubled with depth, **reaching 10 mg L$^{-1}$** but the DOM chromatographic character did not change significantly with depth or along the groundwater flow path but the carbon isotopic composition did change."

p13, l1: Please explain how the estimate of an "export up to ten times" is achieved.
Author's response:
The concentrations found in this coastal system contain up to 10 mg/L of DOC whereas the global medium is 1 mg/L, see above comment.

Figure 2: Perhaps there is a way to improve the quality of the figure (some features appear to be blurred). What do the blue, pink, and red arrows mean in contrast to the black ones?

Agreed. The Figure has been updated and the various coloured arrows have been removed for clarity.

[Figure]

**RC2 -**

**Anonymous Referee #2**

General Comments:

In this paper, Meredith et al. describe the age of groundwater and composition of DOM in a coastal anoxic aquifer in coastal New South Wales. They propose several

DOM sources present within the aquifer, and suggest that anoxic aquifers can contain substantially more DOC than other aquifers and that further research is needed to understand the stabilization of and potential fates of the DOM contained in anoxic aquifers globally. The paper is short and easy to read, and provides important data on anoxic coastal aquifer chemistry to the scientific literature.

Overall, the similarity of DOM composition in the aquifer to the DOM in the wetland seems to need more addressing, particularly given the claim the paper makes that the wetland is not the source of DOM for the aquifer. It would strengthen the paper to further discuss possible reasons (including the possible lack of compositional resolution when using only LC-OCD to assess DOM composition) why the DOM composition is the same, yet the paper claims the sources are be different.

Author's response:

The isotopic differences and the newly added PCA analysis clearly show that the sources of DOC in the aquifer are different. These new lines of evidence are discussed in the final MS. A paper by McDonough et al., (2020b) presenting a different dataset from the site, also shows that the DOM has different ages, with the shallow source close to the wetland reflecting $^{14}C_{DOC}$ ages of ~100 pMC and the deeper samples, or samples further from the wetland, reflecting $^{14}C_{DOC}$ age of ~80 pMC. This is indicative of separate DOM sources in the vicinity of the wetland and for samples further from the wetland or in the deeper groundwater system.

The limits to the compositional resolution of LC-OCD could explain the limited changes observed in the chromatographic character of DOM at the site. High resolution mass spectrometric techniques such as FT-ICR-MS have been used in conjunction with LC-OCD in a paper by McDonough et al., (2020a) at the same study site on a different dataset to further probe changes in groundwater DOC character along the transect. Recent research has demonstrated the utility in identifying groundwater DOC

compositional changes after recharge at the site using LC-OCD and FT-ICR MS (McDonough et al., 2020a).

The two papers by McDonough et al., (2020a and 2020b) are newly published and the work was completed while the current submission was under review with HESS (for over 13 months).

The following analysis will be added to the results section.

page 7 starting line 8

"Principal component analysis (Fig. 6) using water quality parameters, isotopes and DOC variables including LC-OCD fractions (%), $\delta^{13}C_{DOC}$ (‰), $\delta^{13}C_{DIC}$ (‰), $^3$H, pH, Na, $pCO_2$, Cl, Ca, Sr, DO, $SO_4$, $NH_4$, Ca and DOC concentration confirmed the presence of different groundwater sources. The variables contributing to PC1 (in order of importance) were pH, Na, $pCO_2$, Cl, DOC, Ca, humics, LMW-N, Sr and $^3$H. The variables contributing to PC2 in order of importance are $\delta^{13}C_{DOC}$, DO, biopolymers, HS aromaticity, $\delta^{13}C_{DIC}$, HS molecular weight, Cl, DOC and hydrophobic DOM. PC1 mainly explained the variations with sample depth in the aquifer. Samples S1_D, S2_S, S2_D and S3_S were the samples that most strongly influenced PC1. The deep samples likely to have originated from a deeper regional source of water (S1_D and S2_D) were influencing the right hand side of the PCA with high $pCO_2$, humics substances, Na and Cl. These samples were also characterised by low Ca, $^3$H and Sr. The shallow samples (S1_S, S2_S and S3_S) were influencing the left hand side of the PCA with low $pCO_2$, humics, Na and Cl, and have high Ca, $^3$H, and Sr suggestive of rainfall recharge waters. This analysis further highlights the wetland sample that is not related to the other samples (Fig 6) and is heavily influencing PC2 with low $\delta^{13}C_{DOC}$ values, high DO, high biopolymers, low HS aromaticity, low HS molecular weight and high DOC concentration."

Specific Comments:

My main scientific concern about the paper is the claim that DOM is not being processed in these systems. On page 7 line 24-page 8 line 1, a claim is made that DOM would be expected to decrease in DOM aromaticity if it were being biodegraded. I'm not sure this statement holds true for all DOM systems. If DOM were being biodegraded, it could likely become more aromatic along a flowpath due to preferential microbial consumption of more aliphatic, biolabile molecules. I think this statement needs to be backed up by some citations or expanded to explain the degradation pathways being discussed, since higher aromaticity to me suggests preferential microbial processing of biolabile aliphatic fractions is occurring as the DOM moves through the groundwater.

Author's response:

Agreed. This sentence was changed to read:

Page 8, lines 10-12

"The DOC did not show significant or consistent trends in aromaticity or molecular weight, nor did it show a trend of declining DOC concentration with depth or along the flow path".

The MS has been reviewed extensively and all reference to the DOM character being constant has been removed from the MS. It was not the aim of the MS to suggest the DOM is constant. The DOM character varied in the aquifer, but does not along any clear trends hypothesised or expected flowpaths. This just further highlights that there are distinctive sources of DOM in the aquifer and not one wetland source recharging the aquifer and biodegrading along a 500 m transect.

To address the comment that microbial degradation might be expected to break down the aliphatic fraction. We reference the importance of sorption and biodegradation. We also cite Chapelle et al as this paper shows changes in optical clarity along a flowpath that are substantial. We also add the experimental work of Oudone et al., (2019), which confirms that sorption predominantly affects the aromatic, HS fraction. For biodegradation, we cite Shen, Chappelle et al 2014 which shows that only a small proportion of groundwater DOC is bioavailable.

We have rewritten the following sentence to highlight …. "from S3, S4 and S5_S also had higher humic substances aromaticity and humic substances molecular weight than sites 1 and 2." Page 8 line 10

Page 8, ln 20 was added to read "Furthermore, despite observed differences in DOM characteristics (e.g. higher humic substances aromaticity and molecular weight in S3, S4 and S5S), percentages of LC-OCD fractions did not show any consistent trend along the groundwater flow path, contrary to what was found in other studies where biodegradation, sorption, desorption and biosynthesis controlled DOM (Chappelle et al., 2012; Shen et al., 2015).

page 8 starting line 22

"The experimental work of Oudone et al. (2019) also confirmed that sorption predominantly affects the humic substances fraction, especially that with high aromaticity. If biodegradation were occurring, we would expect to see a decline in the LMW-N and BP fractions (Catalán et al., 2017) however we do not make a clear observation of this in our data. In contrast Fig. 6 shows that the shallow samples located further from the wetland (S4_S and S5_S) have higher humic substances aromaticity and molecular weight which supports the conclusion that sorption is not the dominant process determining DOM character from the wetland to the coastline."

Similarly, I do not understand the reasoning presented on page 10 line 6-10. The paper says that the d13C value of DOC should get lighter due to microbial processing, and that yes, in fact the groundwater DOC d13C values are 1 ppm lighter than those in the wetland, yet the paper seems to be saying that this goes against the idea of microbial fractionation. Doesn't that show that yes, the d13C-DOC values are becoming lighter than the original C3 vegetation wetland source, which would support biological processing as a mechanism at work here? I may have missed a piece of logic, but it would strengthen the point of the paper if this were clarified.

Author's response:

Agreed. The following sentence has been added to clarify that the 13C values would be expected to become more enriched with microbial processing. As we only see a 1 ‰ variation in the $^{13}C_{DOC}$ values, we suggest that microbial processing is minimal:

Page 11, line 4-9:

"The carbon isotope value in the wetland was 1 ‰ lighter compared to the shallowest groundwater sample located near the wetland. This suggests a small amount of microbial processing may have occurred, or that unprocessed DOM inputs are very high at this site."

Ln 5, pg 11 "If biological processing was influencing the wetland DOM during transport into the groundwater, it would be expected that the $\delta^{13}C_{DOC}$ values would become heavier than the original C$_3$

vegetation source where the bacteria metabolize the isotopic light organics because the $^{12}$C-H bonds are easier to break leaving the resultant OM more enriched than the original source (Clark and Fritz, 1997)."

We also note:

Page 9, line 19-22: "A possible explanation for the similar constant character of the DOM with the evolved inorganic redox chemistry is that the rates of biodegradation are far lower than the rate of DOM leaching into the groundwater. Furthermore, if there are additional sources of DOM along the flowpath, we would expect to see the inconsistent variations in LC-OCD fractions along the flow path that we observe in our data, rather than a consistent decline in any DOM fraction or DOC concentration."

Further, I do not follow the line of reasoning that claims the modern wetland is not a major source of DOM for these groundwaters on page 10 line 15-16. It appears that DOM composition and d13C are consistent with a wetland source. If the water chemistry is different, it is probable that the DOM source and water source are decoupled, which is interesting. But I do not think the claim that the modern wetland is not the source of DOM to the groundwater is justified based on the explanation of the data provided. Perhaps a more detailed explanation of why the isotopes lead the authors to this conclusion would strengthen the argument.

Author's response:
The aquifer contains several buried peat units that are also the source of DOC for groundwater. We agree the water sources have decoupled from the different to the DOM sources. The new PCA analysis of DOC, isotopes and water chemistry variables now provides the extra reasoning necessary to show that the wetland water is different to the groundwater (Fig 6). In addition, a paper by McDonough et al., (2020b) also shows that the DOM has different ages shallow 100 pmc and deeper source 80 pmc. Page 10, lines 11-14. This paper is newly published and the work was completed while this submission was under review with HESS (for over 13 months). Reference to this recent paper will be made within the final MS discussion.

Other specific comments:
Section 4 is a Results and Discussion section, not a pure Results section, and should be labeled accordingly or take some of the discussion out and put it later in the discussion sectionăĂ˘ I find it a little confusing cause important discussion material is listed here, but then there is a separate discussion section

Author's response:
Agreed. This section has been removed from page 7 lines 1-7 because it is discussed in detail later in the discussion section. This point was also addressed by R1.

Page 7 line 22-24: Was the intention to say that groundwater from sites 3-5 had higher aromaticity than other groundwater/shallower groundwater samples, or higher aromaticity than other surface water samples in other studies? If the former (which is how I initially read this), this seems to contradict the statement in the discussion that

the groundwater DOM composition is constant in this study.

Author's response:

This refers to higher aromaticity and molecular weight at sites 3-5 compared to sites 1 and 2, rather than compared to samples from other studies. We have also changed the statement in the discussion which implied that the groundwater DOM composition is constant in this study. We note that it was not the intention of this study to suggest the DOM is constant. We intend to show that the LC-OCD fractions do not show trends along hypothesised or expected flowpaths. As noted in response to the previous comments, we would expect reduced LMW-N and BP associated with microbial degradation, and a decline in aromaticity and molecular weight associated with sorption, however we do not observe these changes.

Page 11 lines 19-22: It might strengthen the claim that this carbon may be important for the regional carbon budget to explicitly relate this anoxic, preserved DOM to freshly produced DOM, and to explicitly state how it may impact the regional carbon budget.

The claim is made in the conclusion and the abstract that anoxic coastal groundwater systems have the potential to export up to ten times more unreacted carbon than thought, but the calculation or logic to support this is never shown. It would strengthen the claim to demonstrate why that claim is being made, and how the math adds up if the carbon concentrations in the groundwater are only five times higher than those in other aquifers. In general LC-OCD may give too low a resolution to reveal DOM compositional differences between sources and groundwaterâ˘A ˘Tperhaps a discussion of other areas where LC-OCD on its own has been able to tease out compositional differences could strengthen the expectation that if there were compositional differences here, LC-OCD could assess them. Twice it's mentioned that the aromaticity for the humic substances is higher than for lakes or rivers (once in 5.1.2, once in 41.). Especially in the results, it would be useful to see the values from the Huber et al 2011 study and this study to show how they compare.

Author's response:

The concentrations found in this coastal system contain up to 10 mg/L of DOC whereas the global medium is 1 mg/L, hence the emphasise of an order of magnitude difference. The following has been added to the text to clarify:

Page 13, ln 14 "The average groundwater DOC concentration for this study was five times higher (5 mg $L^{-1}$) than the global median DOC concentration for groundwaters. The concentration of DOC doubled with depth, **reaching 10 mg $L^{-1}$** but the DOM chromatographic fractions did not change significantly with depth or along the groundwater flow path. We note that the carbon isotopic composition did however vary slightly."

The groundwater results were originally shown with respect to Huber's study in the old Figure 6 and now in the updated Fig 7. The samples have a higher aromaticity than surface water previously studied. The new work completed by Oudone et al., (2019) including experimental work, clearly shows how LC-OCD can elucidate sorption processes. McDonough et al., 2020a use LC-OCD and FT-ICRMS to understand changes in DOM character associated with adsorption and desorption before and after rainfall recharge. This research shows that changes in DOM fractions due to processing are able to be observed through LC-OCD data. A summary of these works will be included in the final discussion.

Technical Corrections:
Page 4 line 9: "may be" not "maybe"
Author's response: corrected
5  Page 4 line 17: "Dissolved" not "Dissolve"
Author's response: corrected
Page 9 line 5: Organic should not be capitalized
Author's response: corrected
Figure 3: "the polygon" is unclearâ˘Aˇ Tperhaps call it the "dashed rectangle"
10 Author's response: corrected
Figure 6:"Siberian Sea" not "Siberiansea"
Author's response: corrected, the names of the system have been removed on the amended Figure 7.
This is because the systems are not described in detail in the text.

15 Below is the marked-up manuscript version

[revised manuscript text omitted]
 | 20/02/2014 | -23.6 | 18.0 | -30.3 | 1.70 | 0.08 | 0.15 | 4.40 | 1.56 | -3.96 | 0.037 |